# Matrix Metalloproteinase-9 Contributes to Epilepsy Development after Ischemic Stroke in Mice

**DOI:** 10.3390/ijms25020896

**Published:** 2024-01-11

**Authors:** Barbara Pijet, Agnieszka Kostrzewska-Księzyk, Maja Pijet-Kucicka, Leszek Kaczmarek

**Affiliations:** Laboratory of Neurobiology, Braincity, Nencki Institute of Experimental Biology, Pasteura 3, 02-093 Warsaw, Poland; a.kostrzewska@nencki.edu.pl (A.K.-K.);

**Keywords:** ischemic stroke, MMP-9, epileptogenesis, neuronal excitability, gel zymography assays, epilepsy treatments

## Abstract

Epilepsy, a neurological disorder affecting over 50 million individuals globally, is characterized by an enduring predisposition and diverse consequences, both neurobiological and social. Acquired epilepsy, constituting 30% of cases, often results from brain-damaging injuries like ischemic stroke. With one third of epilepsy cases being resistant to existing drugs and without any preventive therapeutics for epileptogenesis, identifying anti-epileptogenic targets is crucial. Stroke being a leading cause of acquired epilepsy, particularly in the elderly, prompts the need for understanding post-stroke epileptogenesis. Despite the challenges in studying stroke-evoked epilepsy in rodents due to poor long-term survival rates, in this presented study the use of an animal care protocol allowed for comprehensive investigation. We highlight the role of matrix metalloproteinase-9 (MMP-9) in post-stroke epileptogenesis, emphasizing MMP-9 involvement in mouse models and its potential as a therapeutic target. Using a focal Middle Cerebral Artery occlusion model, this study demonstrates MMP-9 activation following ischemia, influencing susceptibility to seizures. MMP-9 knockout reduces epileptic features, while overexpression exacerbates them. The findings show that MMP-9 is a key player in post-stroke epileptogenesis, presenting opportunities for future therapies and expanding our understanding of acquired epilepsy.

## 1. Introduction

Epilepsy is a brain disorder characterized by an enduring predisposition to generate epileptic seizures and by the neurobiological, cognitive, psychological, and social consequences of this condition [1,2]. Over 50 million people suffer from the disease, making it one of the most common neurological diseases globally [3]. The occurrence of active epilepsy rises with age, reaching peaks within 5–9 years and beyond 80 years, showing similar patterns in both women and men [1]. In the brain, epilepsy is manifested as structural changes, such as neuronal reorganization, especially prominent within the hippocampus [4,5]. In fact, epilepsy is not a homogenous disorder, but rather a collection of subtypes with a variety of etiologies, including those with a prominent genetic component [6,7]. Acquired epilepsy constitutes about 30% of all cases of epilepsy and is most commonly caused by stroke, brain trauma, alcohol use, neurodegenerative diseases, or infection [8]. In acquired epilepsy, a brain-damaging injury leads to epileptogenesis (latency period without seizures) even lasting up to several years, which culminates in the appearance of seizures and a diagnosis of epilepsy [4]. At present, up to one third of epilepsy cases cannot be controlled by anti-epileptic drugs [9]. Furthermore, there are no therapeutics that prevent epileptogenesis. Therefore, identifying potential anti-epileptogenic drug targets is of paramount importance.

Stroke is the most common cause of acquired epilepsy, especially in the elderly [10,11]. Up to 20% of stroke victims develop epilepsy within months to years after the injury [12,13]. Animal models have greatly contributed to our knowledge of stroke and its deleterious consequences. Nonetheless, studies on stroke-evoked epilepsy are much more scarce. One of the major reasons for this is a poor long-term survival rate of animals, such as mice, following experimental stroke. This is particularly true for the major stroke model, i.e., middle cerebral artery occlusion (MCAO) to produce either transient or permanent ischemia limited to one brain hemisphere. Recently, a novel post-stroke animal care protocol has been developed to overcome the premature death of post-MCAO mice [14], opening a way to investigate the long-term consequences of these injuries, including epileptogenesis.

In the present study, we set out to investigate the role of matrix metalloproteinase-9 (MMP-9) in stroke-evoked epilepsy development. MMP-9 (also known as gelatinase B or type IV collagenase) is a protease that operates mostly extracellularly, to participate in a variety of physiological and pathological phenomena. MMP-9 belongs to a larger group of enzymes, called metzincins, of which over twenty are metalloproteinases [15]. All of these enzymes share structural and functional features and display overlapping substrate specificities. MMP-9 has often been demonstrated to be excessively produced and released in response to injuries that may evoke epileptogenesis [16]. The potential involvement of MMP-9 in epileptogenesis was first suggested based on its activation by seizure-evoking stimuli [17,18]. Next, Wilczynski et al. and Mizoguchi et al. demonstrated the functional involvement of MMP-9 in experimental models of epileptogenesis, employing mice missing the MMP-9 gene that were impaired in developing epilepsy, and rats with additional MMP-9 gene copies, selectively expressed in neurons, which resulted in an enhanced susceptibility to epileptogenesis [19,20]. More recently, Pijet et al. have shown that in a clinically relevant model of animal post-traumatic epileptogenesis, MMP-9 is activated within minutes to hours following brain trauma, and mice missing MMP-9 are somewhat protected, whereas mice with genetically excessive MMP-9 are far more prone to develop epilepsy [21]. Notably, elevated MMP-9 in the epileptic human brain has also been documented [16,22]. Of note, recently, either partially or just poorly selective MMP-9 inhibitors have been shown to offer therapeutic potential towards epileptogenesis [23,24,25,26], further supporting the need to assess a role of MMP-9 in post-stroke epileptogenesis. Our results demonstrate that MMP-9 indeed plays a role in post-stroke epileptogenesis, by virtue of the enzyme activation by MCAO and findings that genetically modified MMP-9 levels (either gene knockout or gene overexpression) affect a seizure threshold following a chemoconvulsant (pentylenetetrazol, PTZ) treatment that is one of the widely recognized measures of susceptibility to develop epilepsy.

## 2. Results

### 2.1. MCAO Produces a Massive Ischemia in the Affected Hemisphere

As an animal model of ischemic stroke, we employed the focal Middle Cerebral Artery occlusion model (MCAO) in mice, as described by Groger et al. in 2005 [27]. For blood flow monitoring, we utilized Laser-Doppler flowmetry, which revealed the moment of occlusion (a 76.66% decrease in blood flow after MCAO compared to the pre-occlusion state; see Figure 1B). To determine the stroke area, we employed TTC (2,3,5-Triphenyltetrazolium) staining 24 h after inducing ischemia. TTC staining revealed an ischemic infarct spanning from the prefrontal cortex through the striatum (milky area; see Figure 1C). To aid in comprehending the experimental results, we have provided the study design in Figure 1D. After undergoing MCAO, mice were allowed 30 days of recovery, after which they were implanted with cortical and hippocampal electrodes for the assessment of neuronal excitability using a PTZ-threshold test under video-EEG monitoring for the next 8 weeks (12 weeks after MCAO).

### 2.2. MCAO Evokes Increases in MMP-9 Levels in the Brain Regions Neighboring the Area of the Injury

In order to describe the temporal pattern of MMP-9 following cerebral ischemia, we obtained brain tissue samples from animals that had undergone the focal Middle Cerebral Artery occlusion model (MCAO) and from those that had undergone sham surgery. The samples were collected at various time points: 10, 30, and 60 min, as well as 2, 6 h, 1, 7, 14, 30, 60, and 90 days post-injury. Tissue samples were collected separately from the ipsilateral cerebral cortex and hippocampus (injured hemisphere) in two groups: MCAO and sham. The sham group underwent surgeries that omitted the artery occlusion step.

Gel zymography was performed to assess MMP-9 levels. Ischemic stroke induced by focal Middle Cerebral Artery occlusion (MCAO) resulted in a significant increase in MMP-9 levels in the cerebral cortex of the ipsilateral hemisphere compared to the sham group. The highest level of MMP-9 activity in MCAO samples was observed 60 min post-occlusion (Figure 2A,B), after which the level of gelatinolytic activity of the MMP-9 enzyme gradually decreased. Within the next 7 days after ischemia, it reached the level of sham-treated animals during the following weeks and months.

### 2.3. MMP-9 Modulates Seizure Threshold Evoked by PTZ in MCAo Mice

Based on our previous study, ten weeks after MCAO mice were implanted with electrodes, including four cortical electrodes and one hippocampal electrode (see Figure 3A) [21]. After one week of recovery, the mice were connected to video-EEG (vEEG) monitoring to perform the PTZ-threshold test. In this test, mice received an i.p. injection of a subthreshold dose of PTZ (30 mg/kg), after which several parameters were described. To evaluate the effect of MMP-9 on post-stroke neuronal excitability changes, we assessed the impact of the injury on the animals’ susceptibility to the subconvulsant injection. To achieve this, we used two strains of genetically modified animals with different levels of MMP-9 gene expression: the first group of animals were MMP-9 knockouts (KO MMP-9) and their wildtype siblings (WT MMP-9). The second group included *mmp-9* gene overexpressing mice (OE MMP-9) and their wildtypes (WT-OE MMP-9) (see Figure 3E–G). All groups had undergone MCAO. Each group had a corresponding control group consisting of their sham-operated littermates. The subdose of PTZ was administered 12 weeks post-MCAO or sham surgery.

In each of the experimental groups, i.e., MMP-9 KO and their respective WT, as well as MMP-9 OE and their WT, the animals subjected to the MCAO procedure displayed significantly reduced latency to the 1st epileptoform spike, indicating enhanced seizure susceptibility as a result of the ischemia. Notably, however, this latency shortening was attenuated in MMP-9 as compared to WT, thus demonstrating that lack of MMP-9 reducing this aspect of the epileptic phenotype. On the other hand, MMP-9 overexpression further diminished the latency, thus potentiating this aspect of epileptogenesis. As far as the number of epileptoform spikes was concerned, this parameter increased as result of MCAO, albeit non significantly in MMP-9 KO, and achieved the highest level in the MMP-9 OE mice. These results also indicated that the lack of MMP-9 diminished the epileptic phenotype, whereas excessive MMP-9 enriched this phenotype.

## 3. Discussion

The major findings of this study can be summarized as follows: Ischemic stroke evokes significant increases in MMP-9 activity in the ipsilateral cortex and hippocampus during first 24 h after the injury. Different levels of *mmp-9* gene expression affect neuronal excitability affected by cerebral ischemia: lack of the active form of the enzyme in MMP-9 KO mice reduces, while overexpression of MMP-9 intensifies susceptibility to the chemoconvulsive agent, PTZ.

Our study shows changes in MMP-9 activity levels using a gel zymography approach, following ischemic stroke induced by focal Middle Cerebral Artery occlusion (MCAOMCAO). Gel zymography is a technique used to visualize the activity of enzymes, in this case, MMP-9. It involves electrophoresis for protein separation based on their molecular weight in a gel and then detects their enzymatic activity towards a substrate filling the gel (gelatin in the case of gelatinases, i.e., MMP-9 and MMP-2). Ischemic stroke was induced by focal MCAOMCAO, a common experimental model for studying stroke in animals. The study employed different time points after the induction of ischemia, specifically looking at changes in MMP-9 activity. We found that ischemic stroke resulted in a significant increase in MMP-9 levels in the cerebral cortex of the ipsilateral hemisphere compared to the sham group. This indicates that ischemic stroke triggers a massive upregulation of MMP-9 activity. The highest level of MMP-9 activity in MCAOMCAO samples was observed 60 min post-occlusion. This suggests a rapid and acute response to ischemic injury, indicating an involvement of MMP-9 in the early stages following the injury. After reaching the peak, the level of gelatinolytic activity of the MMP-9 enzyme gradually decreased. Within the 7 days post-ischemia, MMP-9 activity returned to the level observed in sham animals. The immediate increase in MMP-9 levels post-ischemia (60 min) may suggest an acute response to the ischemic injury, possibly contributing either to early tissue remodeling or damage. This could indicate a role for MMP-9 in the early stages of the response to ischemia, with a subsequent return to baseline levels during the recovery phase. Understanding the temporal dynamics of MMP-9 activity could inform the development of therapeutic interventions targeting MMP-9 to modulate its effects during the different phases of ischemic stroke. Previous studies focused on the time-dependent increases in MMP-9 levels post-injury, demonstrated the changes between the 6 and 5 days after brain ischemia [28]. Moreover, increased MMP-9 activity levels were reported within the first 24 h in both the ipsilateral and contralateral hemisphere [29].

Herein, we have investigated the impact of MMP-9 on post-stroke neuronal excitability, particularly in the context of epileptogenesis. The study showed the changes in neuronal excitability 10 weeks after MCAO, during a subacute phase of stroke recovery. Mice were implanted with electrodes (four cortical electrodes and one hippocampal electrode) for subsequent video-EEG (vEEG) monitoring. The PTZ-threshold test was performed one week after electrode implantation. This test involves administering a subthreshold dose of PTZ (30 mg/kg) to induce epileptiform activity. Two strains of genetically modified animals were used—MMP-9 knockouts (KO MMP-9) and MMP-9 overexpression mice (OE MMP-9). Each experimental group (KO MMP-9 and OE MMP-9) had a corresponding control group consisting of their sham-operated littermates. The KO MMP-9 group displayed a significantly reduced latency to the 1st epileptiform spike compared to their wildtype (WT MMP-9) after MCAO. This suggests enhanced seizure susceptibility. On the other hand, the OE MMP-9 group exhibited further diminished latency, indicating potentiation of epileptogenesis. The number of epileptoform spikes increased as a result of MCAO, although non-significantly in the KO MMP-9 group. The highest level of epileptoform spikes was observed in the MMP-9 OE mice. Those results suggest that the lack of MMP-9 showed a protective effect against enhanced seizure susceptibility, while the MMP-9 overexpression had a potential role in promoting epileptogenesis.

It is interesting to consider whether the seizure phenotype might depend on the stroke volume. The level of MMP-9 activity has been implicated in various vascular and neurological conditions, including its association with stroke volume. Studies have suggested a correlation between increased MMP-9 activity and larger stroke volumes, indicating a potential link between MMP-9 levels and the extent of vascular damage. In fact, we have previously reported that MMP-9 knockout mice have diminished, whereas mice overexpressing MMP-9 have increased brain damage areas, following a traumatic brain injury [21]. Indeed, parameters describing the influence of MMP-9 levels in transgenic mouse groups on neural excitability (latency time, spikes number) suggest that the size of the stroke, dependent on MMP-9 activity levels, may have an impact on epileptic parameters in the PTZ-threshold test.

Within the framework of epilepsy, MMP-9 assumes a diverse role, impacting different facets of the condition. Functioning as an enzyme engaged in the restructuring of the extracellular matrix, MMP-9 is indispensable for processes like tissue repair and inflammation. When activated by an injury in the context of epilepsy, MMP-9 initiates several processes, including the disruption of the blood–brain barrier, the promotion of neuroinflammation, modulation of synaptic plasticity, and the augmentation of neuronal excitability. This study provides insights into the relationship between MMP-9 levels and post-stroke epileptogenesis, suggesting that MMP-9 may play a role in modulating seizure susceptibility after ischemic stroke. Understanding the impact of MMP-9 on post-stroke neuronal excitability could have therapeutic implications, potentially targeting MMP-9 to modulate epileptic outcomes after stroke. The study identifies associations between MMP-9 levels and epileptiform activity. Further experiments are needed to establish causation and understand the underlying mechanisms.

A potentially significant diagnostic aspect involves exploring the correlation between serum MMP-9 levels and epileptic activity in stroke patients. This correlation could serve to identify a potential biomarker for the severity and progression of the disease. Elevated MMP-9 levels, often linked to pathological conditions like post-stroke epilepsy, position it as a valuable biomarker for predicting the development of epilepsy. Tracking MMP-9 levels in serum has the potential to contribute to early detection, prognosis, and evaluation of therapeutic responses across various forms of epilepsy, establishing it as a promising biomarker in clinical diagnostics [30,31,32].

## 4. Materials and Methods

The study involved experiments on mature male mice (aged 12–14 weeks) of the C57BL/6J strain, obtained from Medical University of Bialystok, Center for Experimental Medicine. The analysis was carried out using two transgenic strains: MMP-9 homozygous knock-out mice bred on a C57BL/6J background (referred to as MMP-9 KO mice), along with their corresponding wild-type (WT) siblings [33]. Additionally, mice that over-expressed human pro-MMP-9 under the human PDGF-B promoter were used (referred to as MMP-9 OE mice), also on a C57BL/6J background, along with their respective WT siblings [34]. These strains (C57BL/6J, MMP-9 KO, and MMP-9 OE) were maintained in the Animal House of the Nencki Institute. Before commencing with the experimentation, the mice were housed individually in cages within a controlled laboratory animal facility. The environment was maintained at a temperature of 22 ± 1 °C, with humidity levels ranging from 50% to 60%. The mice had free access to both filtered tap water and standard pelleted chow, and they followed a 12 h light/dark cycle.

### 4.1. Ethics Statement

All procedures were performed in accordance with the EU Directive 2010/63/EU, the ARRIVE guidelines, and pertinent local regulations concerning the welfare of animals used in scientific research. These considerations encompassed the entire spectrum of the study, ranging from its conception and design to the implementation of surgical procedures, post-operative care, and the determination of humane endpoints. Throughout the course of the research, conscientious efforts were made to mitigate animal distress and minimize the necessity for sacrifices. The experiments were approved by the 1st Local Ethics Committee (Permissions Numbers: 514/2018 and 1031/2019).

### 4.2. Focal Middle Cerebral Artery Occlusion Model (MCAOO)

The mice underwent unilateral ischemic stroke using the Middle Cerebral Artery occlusion model. Surgeries were performed according to the MCAOMCAO protocol [27]. Thirty minutes prior to surgery, analgesia was induced via an intraperitoneal (i.p.) injection of butorphanol (3 mg/kg body weight) and next anesthetized with a mixture of ketamine (0.5 g/kg body weight) with medetomidine (0.8 g/kg body weight). The absence of toe-pinch response indicated deep anaesthesia. The skin was be shaved and disinfected with Octenisept (Schulke, Warsaw, Poland). Monitoring of blood flow in the region supplied by the left middle cerebral artery was facilitated using a Laser Doppler Perfusion Monitoring Unit (PeriFlux System 4000, Järfälla, Sweden) and PeriSoft for Windows software (version 2.5.5, Perimed AB, Järfälla, Sweden). The microtip fiber was gently inserted through a small incision in the skin between the left eye and left ear, positioned on the surface of the skull bone directly over the projection of the left middle cerebral artery region and mounted with glue. The second end of the microtip fiber was inserted in the master probe. The mice with the thus entrenched fiber was placed on its back. Then the midline incision was made in the neck and the tissues were spread apart. The left common carotid artery (CCA) was carefully prepared (from surrounding tissue).The first knot was made on the CCA artery using a thread with a diameter of 6.0. Then similar knots were placed on the left external carotid artery (ECA) and the left internal carotid artery (ICA). The blood flow in the internal carotid artery (ICA) was closed using a vascular clip (FST, Fine Science Tools, Heidelberg, Germany). Next, in the left CCA a small hole was made with the use of ophthalmic scissors (FST, Fine Science Tools), after which a silicone nylon-coated filament (diameter with coating 0.19 ± 0.01 mm; 701912PK5Re MCAO suture, Doccol, Sharon, MA, USA) was inserted into the left ICA. Following this, the vascular clip was removed from the artery, and filament was inserted deeper through the internal carotid artery to the middle cerebral artery. The left MCAO induced a rapid drop in blood flow in the left middle cerebral artery blood supply region (around 80% of the pre-MCAO baseline values; Figure 1B). Occlusion lasted for 60 min, after which the filament was removed from the artery. Next, the skin was sutured. To prevent dehydration, the animals were re-administered intraperitoneal 0.9% NaCl (0.5 mL) at the end of the operation. After the surgery, the mice received tolfenamic acid (subcutaneous 4 mg/kg), antisedane stimulant (subcutaneous 1.5 mg/kg), and enrofloxacin (subcutaneous 5 mg/kg). After the surgery, mice were placed in a heated home cage and stayed there until awakening. During the first hours after surgery animals were monitored. In the case of the sham-operated mice, the procedure was similar with the exception of the 60 min filament occlusion. After the filament insertion, it was immediately withdrawn.

### 4.3. Post-Operative Care & Feeding Protocol

The post-operative care period lasted for 1 week [14]. To improve recovery after ischemia, food (standard pellet soaked with water) and water in gel (Vivari; Warsaw, Poland) were provided in a Petri dish within each cage. Additionally, for the five following days animals received i.p. injections of glucose solution (i.p.; 0.5 mL 20%) and physiological 0.9% NaCl (i.p.; 0.5 mL). For next two days after stroke, an analgesic drug (butorphanol, 3 mg/kg b.w.) was s.c. administered to the animals every 12 h. Body weight was measured prior to surgery (i.e., baseline) and daily thereafter until day 7 and then at 14th, 21st, and 30th day post-reperfusion. The change in BW is reported as the percent change relative to baseline. Animal welfare was controlled daily (pain, hypothermia).

### 4.4. Pentyleneterazol (PTZ)-Threshold Test

In accordance with our prior research, we employed the PTZ-threshold test to evaluate the neuronal excitability of the animals [21]. Following electrode implantation, mice received an intraperitoneal injection of a subconvulsant dose (30 mg/kg, i.p.) of PTZ, dissolved in sterile saline. To ensure consistent conditions, this procedure was conducted 10 weeks post-MCAO. For electrode implantation, mice were administered a combination of ketamine and domidor, with dosages matching those detailed in the relevant section. Subsequently, four stainless steel screw electrodes (diameter 1.6 mm, Bilaney Consultants GmbH, Dusseldorf, Germany) were placed. Two recording electrodes were situated above the left and right prefrontal cortex, while a reference and grounding electrode were positioned on the occipital bone above the cerebellum (refer to Figure 3A). The hippocampal electrode position was determined using a mouse brain atlas, specifically at AP—2.0 mm; ML +1.3 mm; DV—1.7 mm from the bregma. These electrode placements were secured with a duracryl mixture. Following surgery, mice were administered tolfine, antisedane, and an antibiotic, and closely monitored until awakening. After an additional 2 week recovery period, mice were individually placed in plexiglass cages and connected to the recording system via commutators (SL6C, Plastics One Inc., Roanoke, VA, USA). A subthreshold dose of pentylenetetrazol (30 mg/kg b.w.) was then injected, and the mice were immediately connected to the monitoring system for a 60 min recording session. The Twin EEG recording system in conjunction with a Comet EEG PLUS featuring a 57-channel amplifier AS40-PLUS (Natus Medical Incorporated, Middleton, WI, USA) facilitated vEEG, which was filtered with a high-pass filter at 0.3 Hz and a low-pass filter at 100 Hz. Simultaneously, the animals’ behavior was captured using a Panasonic I-PRO WV-SC385 digital camera (Tokyo, Japan). Within the first 60 min after PTZ administration, measurements were taken, including latency to the first epileptiform spike, total epileptiform spike count, latency to the initial electrographic seizure, occurrence of seizures (% of animals per group), and mortality (%). Epileptiform spikes were defined as high-amplitude, sharply contoured waveforms with a duration of 20 to 70 ms, exceeding baseline values. Statistical analysis focused on the surviving animals during the 60 min observation period. The total spike count excluded electrographic seizure events. Additionally, we evaluated spontaneous seizures in terms of occurrence, frequency, and duration, with electroencephalographic seizures defined as high-amplitude (>2× baseline) rhythmic discharges persisting for >5 s.

### 4.5. Gel Zymography

Gelatin zymography was executed on cortex and hippocampus tissue samples from both ipsilateral and contralateral sides of MCAOMCAO and sham C57BL/6J mice, employing the methodology outlined by Szklarczyk et al. [18]. Tissue samples were obtained at various time points: 10, 30, and 60 min, as well as 2, 6, and 24 h, 7, 14, 30, 60, and 90 days post-stroke. Brains were extracted and the tissue was dissected on a cold plate before being frozen using dry ice. Each time point group encompassed three animals subjected to MCAO and three sham-operated animals. Samples were stored at −80 °C until analysis. The samples were homogenized in a buffer containing 10 mM CaCl_2_, 0.25% Triton X-100, and a protease inhibitor cocktail (cOmplete mini EDTA-free; Roche, Warsaw, Poland) in water. After centrifugation at 6000× *g* for 30 min, the supernatant containing soluble proteins was collected. The insoluble pellet in Triton X-100 was resuspended in a buffer consisting of 50 mM Tris (pH 7.4), 0.1 M CaCl_2_ in water, and heated at 60 °C for 15 min. Following further centrifugation at 10,000× *g* for 30 min at 4 °C, the pellet was devoid of MMP activities, confirming successful extraction. This Triton X-100-insoluble fraction was established. The entire supernatant post-centrifugation was collected quantitatively. Protein concentration was determined via the BCA protein assay (Pierce, Thermo Fisher Scientific, Waltham, MA USA). Lysed protein samples, devoid of 2-mercaptoethanol, underwent electrophoresis using Tris-glycine 8% acrylamide gels with SDS-PAGE under non-denaturing, non-reducing conditions, containing 0.5% gelatin (Sigma Aldrich, Burlington, MA, USA). Following electrophoresis, gels were washed and incubated in the zymography buffer (50 mM Tris, pH 7.5, 10 mM CaCl_2_, 1 M ZnCl_2_, 1% Triton X-100, and 0.01% sodium azide) for 5 to 7 days. After incubation, gels were stained with 0.5% Coomassie blue G-250 (Sigma Aldrich), and MMP-9 levels were quantified via ImageJ Software (1.53t, Windows version), measuring the optical density of white bands against the blue background.

### 4.6. Statistical Analyses

Statistical analyses were performed using GraphPadPrism 7.0. Changes in MMP-9 activity and neuronal excitability alterations observed in PTZ-threshold test were analyzed with One-way ANOVA with Tukey’s multiple comparisons test. Differences were considered statistically significant if *p* < 0.05.

## 5. Conclusions

In summary, the results presented herein suggest a dynamic and time-dependent regulation of MMP-9 activity in response to ischemic stroke, highlighting its potential significance in the acute and recovery phases of cerebral ischemia. Furthermore, the study provides valuable insights into the role of MMP-9 in post-stroke epileptogenesis, suggesting that its absence may have a protective effect, while its overexpression may potentiate seizure susceptibility. These findings contribute to our understanding of the molecular mechanisms involved in post-stroke complications and may have implications for the development of targeted therapeutic interventions.

## Figures and Tables

**Figure 1 ijms-25-00896-f001:**
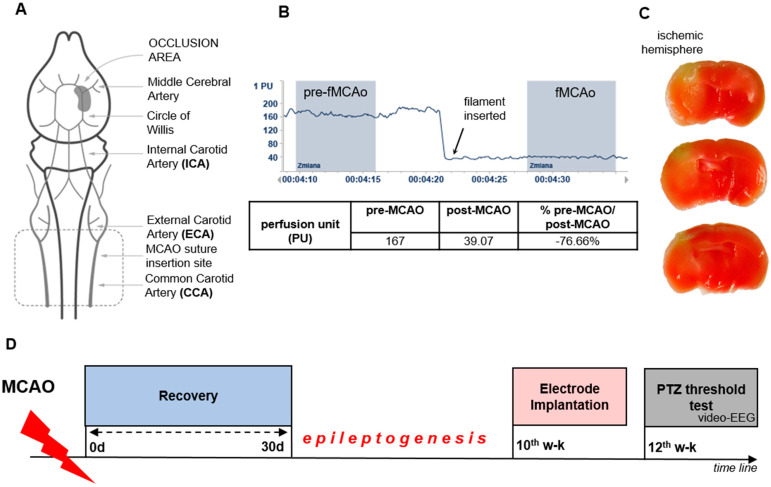
MCAO model and experimental design. (**A**)—Schematic presentation of brain arterial physiology and Middle Cerebral Artery (MCA) occlusion site. Route of the MCAO insertion proceeds from the Common Carotid Artery (CCA) to the occlusion area indicated in grey. The surgery area is located at the bottom of the figure (dotted line). (**B**)—Representative laser Doppler flow; a sharp reduction in cortical blood flow (around 80%) after filament insertion to Internal Carotid Artery (ICA), which occludes the MCA. The occlusion is sustained for the period of 60 min. Infracted area (in white) stained with TTC 24 h after surgery. (**C**)—representative TTC staining sections 24 h after occlusion; (**D**)—Study design; animal condition after surgery was monitored within the first 30 days after ischemia; cortical and hippocampal electrodes were implanted 10 weeks after MCAO; PTZ-threshold test with subthreshold dose of PTZ (30 mg/kg b.w.) was performed 3 months postMCAO (1 h vEEG monitoring).

**Figure 2 ijms-25-00896-f002:**
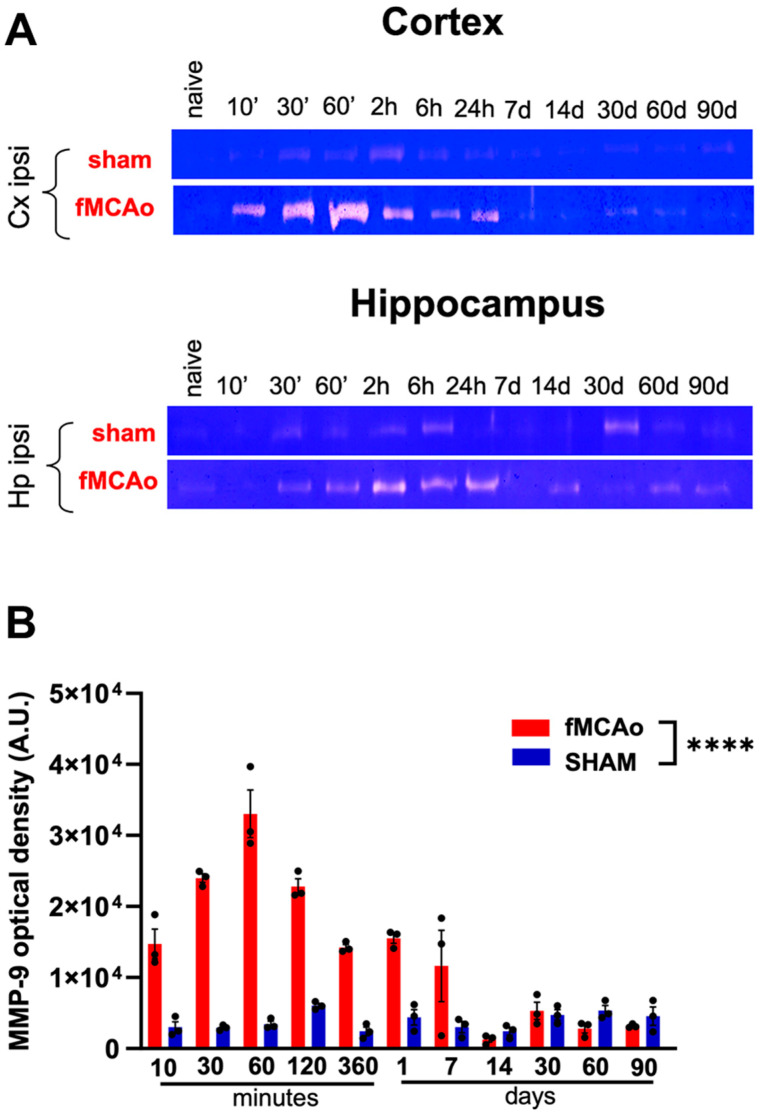
Cerebral ischemia increases MMP-9 activity in ipsi- and contralateral hemisphere. (**A**)—Time-dependent MMP-9 activity in the brain cortex and the hippocampus after MCAO; gel zymography from ipsi- and contralateral cortex (left panel), and the hippocampus (right panel) performed 10, 30, 60 min, 2, 6 h, 7, 14, 30, 60 and 90 days post-MCAO. MCAO mice after focal Middle Cerebral Occlusion model; sham mice (without artery occlusion). Pictures show representative zymograms. (**B**)—Statistical analysis from MMP-9 opitcal density was performed from each time point group (contained 3 MCAO animals and 3 shams; MMP-9 level measured for each sample separately). Data are presented as the mean ± SEM. Statistical analysis was carried out using One-way ANOVA followed by Tuckey post hoc test. **** *p* < 0.0001; data expressed as mean values ± SEM.

**Figure 3 ijms-25-00896-f003:**
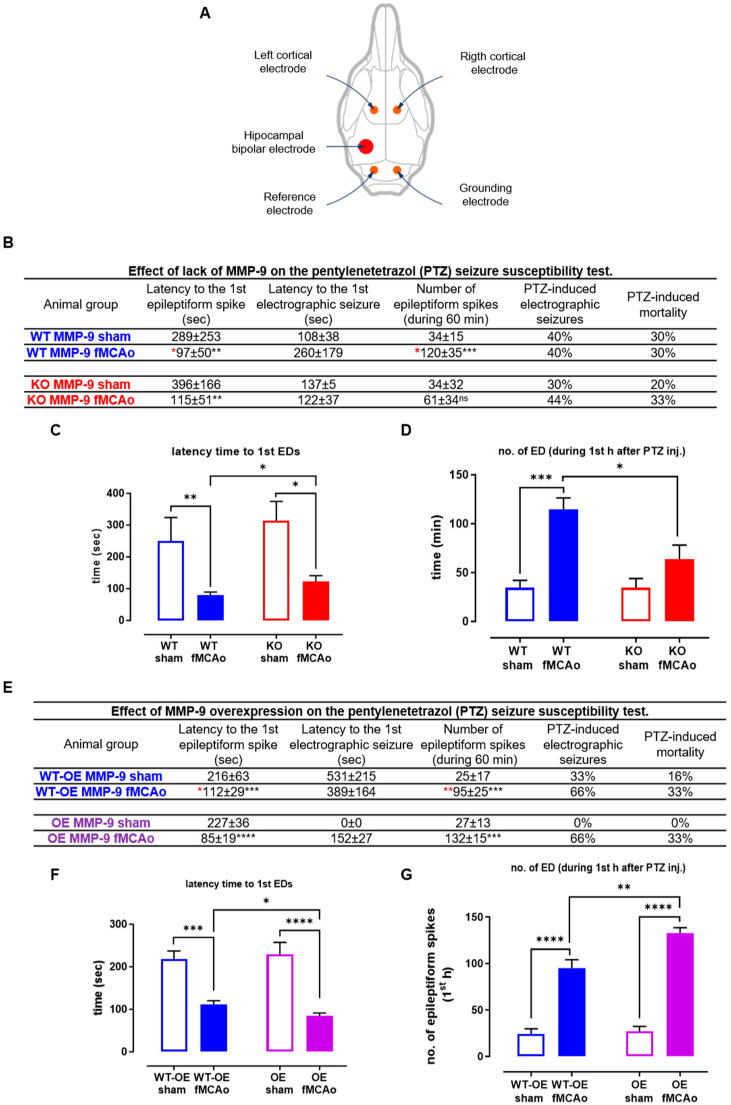
MMP-9 overexpression increases while MMP-9 deficiency diminishes neuronal excitability affected by ischemic stroke. Experiment with use of sub-threshold dose of pentylenetetrazol (PTZ-threshold test) performed 12 weeks after cerebral ischemia. (**A**)—scheme of skull/hippocampal electrodes placement: 2x recording electrodes above the left/right prefrontal cortex, reference/grounding electrodes over the cerebellum, hippocampal electrode position: AP—2.0 mm; ML +1.3 mm; DV −1.7 mm according the bregma; Tables (**B**,**E**) describing the latency time to 1st epileptiform spike (seconds), latency time to the 1st electrographic seizures (seconds) number of epileptiform spikes during 60 min after PTZ injection, % of animals with seizure electographic seizures (%) and PTZ-induced mortality (%); (**B**–**D**)—effect of lack of MMP-9 on neuronal excitability of the animals after ischemic stroke in PTZ-threshold test; (**E**–**G**)—effect of MMP-9 overexpression on neuronal excitability of the animals after ischemic stroke in PTZ-threshold test. Black asterisks indicate statistical significance between the MCAO and sham groups, while red asterisks indicate statistical difference between genotypes. Number of animals in MCAO groups was 9–10, while in sham groups was 5–10. Statistical analysis was carried out using One-way ANOVA followed by Tuckey multiple comparisons post-hoc test * *p* < 0.05; ** *p* < 0.01; *** *p* < 0.001 and **** *p* < 0.0001; ns—not significant; data are expressed as mean values ± SEM.

## Data Availability

Data are contained within the article.

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
