# Peer review of "Matrix Metalloproteinase-9 Contributes to Epilepsy Development after Ischemic Stroke in Mice"

_ijms, 2024, doi:10.3390/ijms25020896_

Round 1

Reviewer 1 Report

Comments and Suggestions for Authors

The manuscript titled “Matrix metalloproteinase-9 contributes to epilepsy development after ischemic stroke in mice” by Pijet, B.; et al.  is a scientific work where the authors study the activity of metalloproteinase-9 (MMP-9) in mice models to observe how the expression of this enzyme triggers the appearance of ischemic strokes and epileptogenesis. The better comprehension of this metalloenzyme could render more effective therapies against the aforementioned diseases which could be extrapolated in humans and could be also useful for other applications like the design of cancer prognosis procedures and the development of quantum technologies.

However, it exists some points that need to be addressed (please, see them below detailed point-by-point) to improve the scientifc quality of the submitted manuscript paper before this article will be consider for its publication in the International Journal of Molecular Sciences.

1) KEYWORDS (OPTIONAL). The authors should consider to add the terms “gel zymography assays” and “epilepsy treatments” in the keyword list.

2) INTRODUCTION. “Epilepsy is a brain disorder characterized (…) Over 50 million people suffer from the disease, making it one of the most common neurological diseases globally” (lines 28-31). It may be convenient if the authors could provide quantitative information about the age and sex prevalence information details of this disease [1].

[1] https://doi.org/10.1016/S1474-4422(18)30454-X

3) RESULTS. It may be advisable if the authors add a schematic representation of the action mechanisms involved in the matrix metalloproteinase-9 and what are the key steps which could trigger epilepsy diseases. Then, does it exist a role of the cellular oxidative stress in the progress of this disease? A brief statement should be provided in this regard.

4) DISCUSSION. “The study provides insights into the relationship between MMP-9 levels and stroke epileptogenesis (…) Further experiments are needed to establish causation and understand the underlying mechanisms” (lines 225-231). Here, the authors need to detail potential applications in the use of metalloproteinase enzymes as their immobilization on sensor devices for the development of quantum technologies [2] or the design of suitable strategies for cancer diagnosis [3].

[2] https://doi.org/10.3390/nano13182585

[3] https://doi.org/10.1039/C8TB02025F

5) MATERIALS & METHODS. “4.6 Nissl Staining and Ischemic Hemisphere Volume Quantification” (lines 356-376). Where are the obtained results related to this section displayed in the main manuscript body text? A figure should be added with representative nissl staining brain sections. Then, the perfusion with paraformaldehyde (line 359) could lead neurotoxicity? A brief discussion should be furnished about this point.

6) CONCLUSIONS. This section summarizes the most relevant outcomes found in this research. Furthermore, the reference citations are in the proper format of the IJMS journal. No actions are requested from the authors.

Comments on the Quality of English Language

The manuscript is well-written. The authors should recheck it in order to polish final details susceptible to be improved.

Author Response

COMMENTS TO AUTHOR:

Reviewer #1:

#1 KEYWORDS (OPTIONAL). The authors should consider to add the terms “gel zymography assays” and “epilepsy treatments” in the keyword list.

Reply: Thank you for your comment; the suggested keywords have been added to the list.

#2 INTRODUCTION. “Epilepsy is a brain disorder characterized (…) Over 50 million people suffer from the disease, making it one of the most common neurological diseases globally” (lines 28-31). It may be convenient if the authors could provide quantitative information about the age and sex prevalence information details of this disease [1].

Reply: Thank you for pointing out this issue; the information has been supplemented in the manuscript.

#3 RESULTS. It may be advisable if the authors add a schematic representation of the action mechanisms involved in the matrix metalloproteinase-9 and what are the key steps which could trigger epilepsy diseases. Then, does it exist a role of the cellular oxidative stress in the progress of this disease? A brief statement should be provided in this regard.

Reply: Thank you for pointing out this issue. We have added a brief description of the mechanisms through which MMP-9 may operate in the development of diseases, including stroke-related seizures (lines 239-245). Certainly, within the realm of insults, the activity of MMP-9 may be impacted by oxidative stress. Conversely, MMP-9 has the potential to contribute to oxidative stress by activating pathways that generate reactive oxygen species (ROS). However, it is important to note that the data presented in the manuscript do not specifically emphasize these relationships.

Now this part reads as follows: Within the framework of epilepsy, MMP-9 assumes a diverse role, impacting different facets of the condition. Functioning as an enzyme engaged in the restructur-ing of the extracellular matrix, MMP-9 is indispensable for processes like tissue repair and inflammation. When activated by an insult in the context of epilepsy, MMP-9 ini-tiates several processes, including the disruption of the blood-brain barrier, the promo-tion of neuroinflammation, modulation of synaptic plasticity, and augmentation of neuronal excitability [30].

#4 DISCUSSION. “The study provides insights into the relationship between MMP-9 levels and stroke epileptogenesis (…) Further experiments are needed to establish causation and understand the underlying mechanisms” (lines 225-231). Here, the authors need to detail potential applications in the use of metalloproteinase enzymes as their immobilization on sensor devices for the development of quantum technologies [2] or the design of suitable strategies for cancer diagnosis [3].

Reply: To follow this point, in the manuscript, a section has been added highlighting the application of diagnostics assessing the MMP-9 levels in the blood serum of patients after ischemic stroke as a biomarker for the development of post-stroke epilepsy.

Now this part reads as follows: A potentially significant diagnostic aspect involves exploring the correlation be-tween serum MMP-9 levels and epileptic activity in stroke patients. This correlation could serve to identify a potential biomarker for the severity and progression of the disease. Elevated MMP-9 levels, often linked to pathological conditions like post-stroke epilepsy, position it as a valuable biomarker for predicting the development of epilep-sy. Tracking MMP-9 levels in serum has the potential to contribute to early detection, prognosis, and evaluation of therapeutic responses across various forms of epilepsy, establishing it as a promising biomarker in clinical diagnostics.

#5 MATERIALS & METHODS. “4.6 Nissl Staining and Ischemic Hemisphere Volume Quantification” (lines 356-376). Where are the obtained results related to this section displayed in the main manuscript body text? A figure should be added with representative nissl staining brain sections. Then, the perfusion with paraformaldehyde (line 359) could lead neurotoxicity? A brief discussion should be furnished about this point.

Reply: Thank you for bringing attention to the above issue. The description of the method for staining brain sections with cresyl violet (Nissl staining) was included inadvertently, and it does not pertain to the results presented in the manuscript. Therefore, it has been removed.

Reviewer 2 Report

Comments and Suggestions for Authors

[IJMS] Manuscript ID: ijms-2788519

Matrix metalloproteinase-9 contributes to epilepsy development after

ischemic stroke in mice

It should be clarified whether the development of seizures was dependend on the stroke-volume.

It should be also reported whether onset of seizures varied between the animals and if this variability was dependent on the amount of MMP-0

12/23

Comments on the Quality of English Language

minro revision requested

Author Response

COMMENTS TO AUTHOR:

Reviewer #2

#1 It should be clarified whether the development of seizures was dependend on the stroke-volume.

Reply: Thank you for pointing out this important issue. To follow the suggestions, the appropriate comments has been added to the Discussion in the revised ms.

Now this part reads as follows: It is interesting to consider whether the seizure phenotype might depend on the stroke-volume. The level of MMP-9 activity has been implicated in various vascular and neurological conditions, including its association with stroke volume. Studies have suggested a correlation between increased MMP-9 activity and larger stroke volumes, indicating a potential link between MMP-9 levels and the extent of vascular damage. In fact, we have previously reported that MMP-9 knockout mice have diminished, whereas mice overexpressing MMP-9 have increased brain damage area following traumatic brain injury [21]. Indeed, parameters describing the influence of MMP-9 lev-els in transgenic mouse groups on neural excitability (latency time, spikes number) suggest that the size of the stroke, dependent on MMP-9 activity levels, may have an impact on epileptic parameters in the PTZ-threshold test.

#2 It should be also reported whether onset of seizures varied between the animals and if this variability was dependent on the amount of MMP-9

Reply: In the fMCAo model, we did not observe spontaneous seizures; therefore, speculation regarding seizure onset and its MMP-9 dependent variability was not addressed in the above manuscript. Our observations are solely based on the results from the pentylenetetrazole test, where seizures were induced using a sub-threshold dose of the convulsant.
